# Innovative Neuroimaging Biomarker Distinction of Major Depressive Disorder and Bipolar Disorder through Structural Connectome Analysis and Machine Learning Models

**DOI:** 10.3390/diagnostics14040389

**Published:** 2024-02-10

**Authors:** Yang Huang, Jingbo Zhang, Kewei He, Xue Mo, Renqiang Yu, Jing Min, Tong Zhu, Yunfeng Ma, Xiangqian He, Fajin Lv, Du Lei, Mengqi Liu

**Affiliations:** 1Department of Radiology, The First Affiliated Hospital of Chongqing Medical University, Chongqing 400016, China; 2College of Medical Informatics, Chongqing Medical University, Chongqing 400016, Chinamichellemin42@gmail.com (J.M.);

**Keywords:** bipolar disorder, major depressive disorder, gray matter, graph theory, machine learning

## Abstract

Major depressive disorder (MDD) and bipolar disorder (BD) share clinical features, which complicates their differentiation in clinical settings. This study proposes an innovative approach that integrates structural connectome analysis with machine learning models to discern individuals with MDD from individuals with BD. High-resolution MRI images were obtained from individuals diagnosed with MDD or BD and from HCs. Structural connectomes were constructed to represent the complex interplay of brain regions using advanced graph theory techniques. Machine learning models were employed to discern unique connectivity patterns associated with MDD and BD. At the global level, both BD and MDD patients exhibited increased small-worldness compared to the HC group. At the nodal level, patients with BD and MDD showed common differences in nodal parameters primarily in the right amygdala and the right parahippocampal gyrus when compared with HCs. Distinctive differences were found mainly in prefrontal regions for BD, whereas MDD was characterized by abnormalities in the left thalamus and default mode network. Additionally, the BD group demonstrated altered nodal parameters predominantly in the fronto-limbic network when compared with the MDD group. Moreover, the application of machine learning models utilizing structural brain parameters demonstrated an impressive 90.3% accuracy in distinguishing individuals with BD from individuals with MDD. These findings demonstrate that combined structural connectome and machine learning enhance diagnostic accuracy and may contribute valuable insights to the understanding of the distinctive neurobiological signatures of these psychiatric disorders.

## 1. Introduction

Bipolar disorder (BD) is a prevalent and debilitating mental health condition marked by recurrent episodes of mania and depression [1]. The clinical symptoms of BD overlap with those of other psychiatric disorders, such as major depressive disorder (MDD), schizophrenia (SZ), and attention deficit hyperactivity disorder (ADHD) [2], which can lead to misdiagnosis. Among them, MDD is the most common misdiagnosis for BD. MDD is a widespread psychiatric condition characterized by persistent low mood, diminished interest or pleasure, and other significant cognitive and physical symptoms [3]. Distinguishing between MDD and BD in clinical practice poses a significant challenge due to the overlap in their symptoms. Currently, the differential diagnosis between BD and MDD relies primarily on clinical symptoms, which are inherently subjective and lack sufficient differential accuracy. Previous studies indicated that as many as 69% of BD patients have been misdiagnosed [4,5,6]. This diagnostic ambiguity can lead to delays in appropriate treatment initiation and hinder the development of targeted therapeutic strategies tailored to the specific needs of each disorder [7]. Hence, gaining a comprehensive understanding of the pathophysiology of BD and MDD, particularly, their distinctions, is of paramount importance.

The utilization of structural imaging through MRI has proven invaluable not only in comprehending the biology of mental illnesses but also in assessing the impact of medications and predicting treatment responses [3,8]. While anatomical structures were traditionally considered relatively static in adult life, recent findings highlight measurable alterations in gray matter structures resulting from behavioral training and acute psychotropic medication treatments. Noteworthy studies include those exploring antipsychotics and lithium effects in individuals with MDD and BD [9,10,11]. Furthermore, accumulating evidence points to network dysconnectivity as a key aspect of MDD and BD pathophysiology [12,13,14]. Graph-based analysis stands out as a powerful tool for characterizing the topological properties of brain networks, known as the connectome [15,16], and for elucidating alterations in psychiatric disorders [17,18]. For example, recent reports highlighted the identification of potential causes for the alteration of the connectome structure in BD [19,20] and MDD [21,22] through graph theory analysis. Therefore, investigating the brain connectome presents a promising strategy for advancing our understanding of these illnesses and developing biomarkers. Recent advancements in high-resolution structural MRI have facilitated the delineation of whole-brain connectivity patterns by evaluating interregional gray matter (GM) volume measurements [23,24,25,26]. Such measures play a crucial role in understanding how changes in network anatomy influence functional and behavioral characteristics [27,28]. In the context of our research, these sophisticated techniques offer a comprehensive means to explore and differentiate structural connectome alterations between MDD and BD individuals, contributing to the development of a robust diagnostic framework.

In recent years, the integration of machine learning into psychiatric research has shown promise for unraveling complex diagnostic puzzles. Various studies successfully employed machine learning algorithms to discriminate between different mental health conditions, often outperforming traditional diagnostic methods. Specifically, in the realm of BD and MDD, where clinical differentiation is challenging, machine learning has emerged as a potent tool. Previous research demonstrated the feasibility of utilizing machine learning models for predicting treatment responses, evaluating medication effects, and distinguishing between diverse psychiatric disorders based on neuroimaging data [3,8,29]. By leveraging intricate brain connectivity patterns and employing advanced graph theory methodologies, our research aims to unveil distinct neurobiological signatures. This integration has the potential to revolutionize diagnostics, providing clinicians with an objective, data-driven method for distinguishing between these psychiatric disorders. The anticipated findings not only promise to enhance clinical decision making but also offer valuable insights into the underlying neural mechanisms differentiating BD from MDD. Through this innovative fusion of neuroimaging, graph theory, and machine learning, our study aspires to improve the differentiation and understanding of major depressive disorder and bipolar disorder in clinical practice.

In this study, we propose the integration of structural connectome analysis with machine learning models to discern distinctive patterns in these disorders. We aimed to combine graph theory analysis and machine learning models to distinguish between MDD and BD, thereby assisting clinical diagnosis. First, we anticipated distinct graph theoretical attributes in both MDD and BD compared to healthy controls and between the two disorders, and second, we expected that the integration of structural connectome and machine learning models would effectively differentiate between MDD and BD. This research contributes to advancing diagnostic precision for psychiatric disorders, offering potential insights for enhancing clinical decision making in differentiating between MDD and BD.

## 2. Materials and Methods

### 2.1. Participants

A total of 75 consecutive inpatients (first-episode BD, BD group, *n* = 32; first-episode unipolar depression, MDD group, *n* = 43) and 44 healthy controls (HC group) were enrolled from The First Affiliated Hospital of Chongqing Medical University, Chongqing, China. All patients were diagnosed based on the Diagnostic and Statistical Manual 5 (DSM-5) Structured Clinical Interview (SCID) by two professional psychiatrists independently. In this study, 4 out of 32 patients in the BD group and 4 out of 43 patients in the MDD group were drug-naïve, while the remaining patients received treatment with various antipsychotics, antidepressants, and mood stabilizers prior to the present study. The demographic characteristics, including age, sex, education, duration of the illness, and use of psychotropic medications, were collected from all participants. The 24-item Hamilton Depression Rating Scale (HAMD), 14-item Hamilton Anxiety Scale (HAMA), and Young Mania Rating Scale (YMRS) [30] were applied to assess the severity of the symptoms. Inclusion criteria for all BD and MDD patients included the following: (1) the patients met the diagnostic criteria according to the DSM-5; (2) the first episode was diagnosed; (3) the patients were right-handed. HCs were required to have an HAMD-24 total score ≤7 and no family history of psychiatric disorders in their first-degree relatives. Exclusion criteria for all participants included the following: (1) the presence or a history of head trauma or of neurological or comorbid psychiatric disorders; (2) substance abuse or dependence; (3) a history of electroconvulsive therapy, transcranial magnetic stimulation, and psychotherapy; (4) current pregnancy or breastfeeding; (5) any conditions not suitable for MRI scanning. The study was conducted following the guidelines of the Helsinki Declaration and was approved by the research ethics committee of Chongqing Medical University. Written informed consent was obtained from all participants and/or their guardians.

### 2.2. Data Acquisition

MRI data were acquired on a 3.0-T MRI system (Skyra, Siemens Healthcare, Erlangen, Germany) with a 32-channel head coil. Foam pads and earplugs were used to minimize head motion and reduce MRI scanner noise for each participant. Conventional axial T2-weighted images and fluid-attenuated inversion recovery images with 5 mm slice thickness were first acquired to exclude obvious structural abnormalities. Then, high-resolution T1-weighted structural images were acquired using a three-dimensional volumetric magnetization-prepared rapid gradient−echo (MPRAGE) sequence. The corresponding imaging parameters were repetition time (TR) = 2000 ms, echo time (TE) = 2.56 ms, inversion time (TI) = 900 ms, flip angle = 9°, matrix size = 256 × 256, field of view (FOV) = 256 mm × 256 mm, slice thickness = 1 mm, slices per slab = 192, and voxel size = 1.0 mm × 1.0 mm × 1.0 mm. Finally, all images were visually inspected by two senior radiologists to ensure that no gross lesions or obvious scanning artifacts were present.

### 2.3. MRI Data Preprocessing

The Statistical Parametric Mapping software (SPM12; http://www.fil.ion.ucl.ac.uk/spm, accessed on 3 February 2024) was used to process the structural images. Initially, individual structural images were segmented into gray matter (GM), white matter (WM), and cerebrospinal fluid (CSF) using the unified segmentation model [31]. Subsequently, the resulting GM maps underwent high-dimensional “diffeomorphic anatomical registration through exponentiated Lie algebra (DARTEL)” normalization to the Montreal Neurological Institute (MNI) space and nonlinear modulation to compensate for spatial normalization effects. Finally, the GM data were resampled to 2 mm^3^ voxels and subjected to spatial smoothing using a Gaussian kernel with a full width at half maximum of 6 mm.

### 2.4. Construction of GM Structural Networks

We utilized the automated anatomical labeling (AAL) algorithm to partition the brain’s GM into 90 non-cerebellar anatomical regions of interest (ROIs) [32]. The edges, representing morphological connections, were defined based on the statistical similarity of morphological distributions between different AAL-defined regions. This similarity was quantified using the Kullback–Leibler divergence-based similarity (KLS) method as described by Kong et al. [33,34]. To construct the connection matrices for each subject’s network, we extracted the GM intensity values from all voxels within each ROI. Subsequently, we estimated the probability density function (PDF) of the GM intensity values for each ROI using kernel density estimation (KDE).

Kullback–Leibler divergence is a probability theory measure used to quantify the difference between two probability distributions or the amount of information lost when approximating one distribution to another. In our study, we computed the KLS values for all possible pairs of the 90 brain regions, obtaining a 90 × 90 similarity matrix for each subject. In this matrix, each row and column corresponded to a specific brain region, and each element represented the morphological distribution similarity between the brain regions. The range of KLS values is from 0 to 1, with 1 indicating an identical distribution between two regions. By estimating the similarity of regional morphological distributions, we were able to capture the interregional anatomical similarities within each participant while considering the complex structure of the cerebral cortex. This approach enhanced our ability to characterize the morphological network topology.

### 2.5. Network Properties

We utilized the GRETNA toolbox version 2.0 (http://www.nitrc.org/projects/gretna/, c) to calculate various network properties of brain networks. To ensure that the thresholded networks were estimable for the small-worldness scalar and that the small-world index (σ) was greater than 1.0 [35], we applied a broad range of sparsity (S) thresholds to all correlation matrices. The S value was carefully selected for each threshold to ensure that the resulting networks could be analyzed. The range of S thresholds was set from 0.10 to 0.34, with intervals of 0.01. For each network metric, we calculated the area under the curve (AUC), which provides a summary scalar for the topological characterization of brain networks. The AUC metric is sensitive in detecting topological alterations of brain networks [36,37]

At each sparsity threshold, we computed both global and nodal network properties for brain networks. Global metrics, including small-world [35] and network efficiency parameters [30], were examined. The small-world parameters comprised clustering coefficient (Cp), characteristic path length (Lp), normalized clustering coefficient (γ), normalized characteristic path length (λ), and small-worldness (σ). The network efficiency parameters included local efficiency (Eloc) and global efficiency (Eglob). Additionally, we examined metrics pertaining to individual nodes, such as nodal degree, nodal efficiency, and betweenness centrality [37,38].

### 2.6. Statistical Analysis

To identify significant differences in network properties between groups, nonparametric permutation tests [37] were employed on the AUC of each network metric. To examine null hypotheses, all values for each network metric were randomly reassigned to two groups, and mean differences between them were recalculated. This randomization process was repeated 10,000 times, and the 95th percentile points of each distribution were used as critical values for a two-tailed test of the null hypothesis with a type I error rate of 0.05. Additionally, the false discovery rate (FDR) correction was applied to assess differences in nodal measures.

Independent sample *t*-tests were used in SPSS software (http://www.spss.com), version 27.0, to compare clinical indicators and global properties of network parameters between MDD and BD. Analysis of variance (ANOVA) with the least-significant-difference (LSD) method was used to test for differences in age among the HC, BD, and MDD groups. Sex was examined using the chi-square test.

To examine the relationship between the network parameters of MDD and BD with the HAMD and HAMA, partial correlation analysis was conducted, with age and sex as covariates. Additionally, the relationship between the network parameters of BD and the Young Mania Rating Scale was also examined. The obtained *p*-values were corrected using the false discovery rate (FDR) correction (*q* = 0.05). Statistical analysis was conducted using R software (version 4.1.2), and the R package “ppcor” (https://cran.r-project.org/web/packages/ppcor/index.html) was employed for partial correlation analysis.

### 2.7. Machine Learning Models

We attempted to differentiate control subjects from those with BD and MDD. Each subject’s GM morphological matrices (90 × 90 Pearson correlation matrices) were used as features. These were then input into the same nine machine learning models mentioned above for training. We employed three common machine learning models in MRI imaging: support vector machine (SVM), random forest (RF) [39,40], and eXtreme gradient boosting (Xgboost) [39,41]. Additionally, we utilized common deep learning dimensionality reduction algorithms, such as autoencoder (AE) and deep neural network (DNN), to reduce the dimensionality of the morphological matrices. Similar approaches were previously documented in the literature with favorable classification outcomes [34,42]. The reduced features were then input into the aforementioned three common machine learning models for classification.

Using the GM morphological matrices (90 × 90 Pearson correlation matrices) of each subject in the BD and MDD groups as feature inputs, we attempted to differentiate between MDD and BD. We trained SVM, RF, and Xgboost models. Additionally, we employed AE and DNN for feature reduction, and subsequently trained the aforementioned three conventional machine learning models using the reduced features.

We extracted corresponding morphological submatrices (15 × 15 Pearson correlation matrices) from brain regions that showed differences between BD and MDD based on statistical analysis and used them as feature inputs for the SVM, RF, and Xgboost models for classification. Furthermore, we attempted to perform effective classification by reducing the dimensionality of the features using autoencoder and deep neural network and then inputting the reduced features into the three models.

To mitigate the confounding effect of medication on the machine learning models differentiating between BD and MDD, we coded no medication use as 1, the use of antidepressants alone as 2, and the combined use of antidepressants and antipsychotic drugs as 3. Because of the non-linear relationship between features and labels, we employed the Gaussian process regression method along with a kernel function to control for medication status. Specifically, a regression model was established for each feature to capture how it varied with the medication status, and the predicted feature values were subtracted from the actual feature values to obtain residuals. These residuals were subsequently used as features in the analysis. Additionally, the features were standardized and normalized after controlling for confounding variables.

All models were trained and evaluated using 5-fold stratified cross-validation. For each iteration of cross-validation, a subset of the dataset was used to evaluate the model (i.e., test set), while the remaining four subsets were used for training (i.e., training set). In the dimensionality reduction stage, we only performed dimensionality reduction based on the training set, and the test set was solely used for performance evaluation, never for model tuning or training. Model performance was determined by balanced accuracy, sensitivity, and specificity. Permutation testing was also employed to validate the models. By shuffling the data labels 10,000 times (for instance, exchanging labels of some data in a binary classification problem), retraining the model, and evaluating its performance, we could assess whether the model’s predictions significantly surpassed random guessing.

## 3. Results

### 3.1. Demographic and Clinical Characteristics

There were no statistically significant differences in age and sex among the BD, MDD, and HC individuals enrolled. Similarly, no significant differences were found in the HAMD and HAMA scores between the MDD and the BD patients (Table 1).

### 3.2. Alterations in Global Brain Network Properties

In the defined threshold range, there was no global property difference between MDD and BD patients in the small-world topology of brain structural connectomes. However, compared to the HC group, the BD group exhibited significantly increased values of Eglob (*p* = 0.022), Eloc (*p* = 0.046), and σ (*p* = 0.013). Similarly, the MDD group showed elevated values of Eglob (*p* = 0.043), Eloc (*p* = 0.030), and σ (*p* = 0.014) when compared to the HC group (Figure 1).

### 3.3. Alterations in Nodal Brain Network Properties

Multiple comparison correction using the FDR threshold q = 0.05 was applied to nodal centrality analysis. Common differential brain regions in MDD and BD patients compared to HCs included the right parahippocampal gyrus, the right amygdala, the right superior occipital gyrus, and the bilateral superior parietal gyrus.

In a comparison between BD and HC subjects, unique differential brain regions included the bilateral superior frontal gyrus, dorsolateral; the bilateral middle frontal gyrus; the right inferior frontal gyrus, triangular part; the right inferior frontal gyrus, orbital part; the left Rolandic operculum; the bilateral supplementary motor area; the bilateral superior frontal gyrus, medial; the left amygdala; the right postcentral gyrus; the left paracentral lobule; and the right temporal pole, superior temporal gyrus.

Moreover, in a comparison between MDD and HC subjects, the unique differential brain regions included the right inferior frontal gyrus, the right superior frontal gyrus, the medial orbital, the right anterior cingulate and paracingulate gyri, the right posterior cingulate gyrus, the left hippocampus, the right angular gyrus, the right precuneus, the left thalamus, and the left Heschl gyrus (Table 2 and Figure 2).

### 3.4. Correlation of Network Alterations with Clinical Symptom Severity

Age and sex were included as covariates in the partial correlation analysis. When comparing the BD group to the HC group, we found a significant negative correlation between HAMD scores, HAMA scores, and nodal efficiency (HAMD scores, *r* = −0.496, *p* = 0.008, FDR-corrected; HAMA scores, *r* = −0.430, *p* = 0.025, FDR-corrected) and the degree (HAMD scores, *r* = −0.520, *p* = 0.005, FDR-corrected; HAMA scores, *r* = −0.568, *p* = 0.002, FDR-corrected) of the right middle frontal gyrus (Figure 3). Furthermore, for the MDD group, the HAMD scores (*r* = −0.425, *p* = 0.006, FDR-corrected) and HAMA scores (*r* = −0.383, *p* = 0.014, FDR-corrected) demonstrated a significant negative correlation with Eglob, a global metric (Figure 3). However, we did not observe any significant correlation between the YMRS scores and the network parameters in BD patients.

### 3.5. Classification Performance

Utilizing GM morphological matrices (90 × 90) as features to differentiate HCs from patients with BD and MDD, the mean balanced accuracy for classification was 75.1% using SVM, 87.9% for RF, and 91.1% for Xgboost. Autoencoder was applied as the first-stage dimensionality reduction technique, leading to mean balanced accuracy percentages of 67.0% for SVM, 85.7% for RF, and 86.7% for Xgboost in the second stage. When DNN was utilized as the first-stage dimensionality reduction technique, the mean balanced accuracy in the second stage was 63.5% for SVM, 83.5% for RF, and 83.2% for Xgboost (Table 3).

When utilizing the GM morphological matrix (90 × 90), the mean balanced accuracy for classifying MDD and BD was 78.8% for SVM, 83.7% for RF, and 85.0% for Xgboost. Employing autoencoder as the first-stage dimensionality reduction technique followed by SVM, RF, and Xgboost as the second-stage dimensionality reduction techniques yielded mean balanced accuracies of 79.2%, 80.2%, and 79.7% for SVM, RF, and Xgboost, respectively. Meanwhile, utilizing DNN as the first-stage dimensionality reduction technique resulted in mean balanced accuracy percentages of 57.3% for SVM, 78.0% for RF, and 76.0% for Xgboost in the second stage (Table 3).

By employing the submatrix of the GM morphological matrix (15 × 15), the mean balanced accuracy for classification was 81.0% using SVM, 84.3% for RF, and 87.0% for Xgboost. Autoencoder was applied as the first-stage dimensionality reduction technique, leading to mean balanced accuracy percentages of 73.8% for SVM, 85.5% for RF, and 83.0% for Xgboost in the second stage. When DNN was utilized as the first-stage dimensionality reduction technique, the mean balanced accuracy in the second stage was 86.8% for SVM, 83.3% for RF, and 90.3% for Xgboost (Table 3).

## 4. Discussion

In our study, we constructed individual whole-brain GM morphological networks using structural MRI data for patients with BD and MDD. Employing graph theory-based analytics, we identified significant alterations in structural connectome metrics, highlighting distinct differences between MDD and BD patients compared to HCs. These findings underscore disrupted network efficiency in both MDD and BD, as well as unique differences between the two psychiatric conditions. Furthermore, utilizing machine learning techniques, we achieved a 90.3% accuracy in distinguishing between the BD and the MDD groups based on these structural brain parameters. This dual approach of graph theory and machine learning offers valuable insights into the neuroanatomic distinctions between BD and MDD, demonstrating potential for enhanced diagnostic precision in psychiatric disorders.

Although the small-world property was found in the GM morphological networks of all three groups, we observed significant inter-group differences in several important small-world metrics and in network efficiency. In the present study, the global properties did not differ between the BD group and the MDD group. However, both BD and MDD patients exhibited increased Eglob, Eloc, and σ compared to the HC group, suggesting that brain GM networks of both BD and MDD patients may develop towards stronger small-world properties. Eglob is considered to indicate the level of global integration of a network, whereas Eloc represents the level of local functional segregation [38]. The combination of high local and global efficiencies is thought to support information processing and mental representations through segregated and integrated information processing [15]. The BD and MDD GM networks were characterized by higher integration and higher segregation, reflected by greater Eglob and Eloc, compared with the control networks. Our findings are compatible with previous studies [43,44]. Furthermore, increased small-world organization of other types of brain networks, such as functional connectivity networks [37] or white matter structural connectivity networks [45], was also reported. Thus, our findings further demonstrate that BD and MDD are disorders with disturbance of brain networks. On the contrary, global topological alterations in small-world properties are controversial. Several studies similar in scale reported decreased Eglob and Eloc [46,47] or no significant alterations [48]. These inconsistencies may be due to several factors, including sample heterogeneity (age, course of the disease, different number of episodes), the use of different modalities (structural or functional images), or different constructions of the network matrix (network sparsity densities and graph threshold).

At the regional level, we found that patients with BD and MDD showed common differences in nodal parameters predominantly in the right amygdala and the right parahippocampal gyrus when compared with HCs. The amygdala and parahippocampal gyrus are part of the limbic and paralimbic system and are regarded as the core components of the emotional regulation circuit [49]. Previous studies reported multiple changes in the structure and function of these regions in patients with mood disorders [50,51,52,53,54], suggesting that these structures are involved in the pathophysiological mechanisms of mood disorders. In addition, compared with HCs, distinctive differences were found in BD patients mainly in prefrontal regions (e.g., bilateral dorsolateral prefrontal cortex, bilateral middle frontal gyrus, and bilateral medial frontal gyrus), whereas MDD patients were characterized by abnormalities predominantly in the left thalamus and default mode network. Moreover, when compared directly with the MDD group, the BD group demonstrated altered nodal parameters predominantly in the fronto-limbic network (e.g., bilateral dorsolateral prefrontal cortex, bilateral middle frontal gyrus, bilateral medial frontal gyrus, bilateral supplementary motor area, left hippocampus, and left thalamus). The prefrontal cortex is responsible for processing higher-order executive functions, whereas the thalamus is a core hub that links the limbic network (amygdala, striatum, and hippocampus) to multiple areas of the prefrontal cortex [55]. Therefore, the thalamus integrates neural signals to maintain normal and efficient activities of the fronto-limbic circuit. As a result, dysfunction of the thalamus may lead to “dissociation” of the activity of the circuit. A series of previous studies compared differences in brain structure and function between BD and MDD, revealing multiple alterations in the fronto-limbic circuit [56,57,58]. These findings further support the crucial role of the fronto-limbic circuit in distinguishing between BD and MDD.

It is worth noting that significant negative correlations were observed between the severity of the symptoms (HAMD and HAMA scores) and the nodal centrality of the right middle frontal gyrus in both BD and MDD patients in our sample. The absolute value of the correlation coefficient was only moderate, but it still holds certain clinical significance. The middle frontal gyrus, which plays a crucial role in advanced cognitive processes such as decision making and integration of emotion and information [59], has been extensively reported to present morphological as well as functional alterations in patients with BD and MDD [60,61]. Notably, amplitude of low-frequency fluctuations in the right middle frontal gyrus exhibited a negative correlation with cognitive task performance in individuals with euthymic BD [60]. Therefore, dysfunction of the middle frontal gyrus may be associated with cognitive decline and potentially contribute to the pathophysiology of mood disorders. These findings suggest that nodal centrality of the right middle frontal gyrus holds promise for predicting symptom severity among patients with mood disorders.

In clinical practice, the presence of mania is considered the main difference between BD and MDD. In addition, certain clinical features can help distinguish BD from MDD. Patients with BD tend to be characterized by a lower female-to-male ratio, earlier onset age, increased recurrence, and poorer response to antidepressants [62,63]. Therefore, various methods have been tried to improve the accuracy of differential diagnosis, and one of the most important methods is machine learning. Among machine learning studies, SVM was the most commonly used machine learning model in diagnosing BD, followed by ANNs, ensemble models, linear regression, and the Gaussian process model [64]. The diagnostic accuracy of these studies ranged from 64% to 98%. In our study, we trained 18 models using two different inputs, i.e., the GM morphological matrix and the submatrix of the GM morphological matrix, and the combined Xgboost and DNN model, with the GM submatrix as input, was ultimately established to differentiate BD from MDD, achieving an accuracy of 90.3%. The model metrics indicated that the overall accuracy of the models using the GM shape matrix as input was lower than that of models using the GM shape submatrix. Moreover, traditional machine learning models had lower accuracy compared to machine learning models that utilized deep learning for dimensionality reduction. In our dataset, the GM shape matrix was a 90 × 90 matrix, whereas the GM shape submatrix was a 15 × 15 matrix. The lower dimensionality of the 15 × 15 submatrix makes it easier for machine learning models to identify distinct classification features compared to higher-dimensional matrices like the 90*90 one. Deep learning models, such as DNN and AE, automatically and effectively address complex nonlinear problems by representing high-dimensional data through deep layer modeling [65]. This enables the extraction of the most prominent features from the data, leading to improved classification performance [66]. However, in the distinction of control subjects from those with BD and MDD, we found that Xgboost without dimensionality reduction achieved the highest classification accuracy (91.1%), indicating that the selection of machine learning models should be based on actual data situations. Nevertheless, it is worth noting that we obtained a relatively high classification accuracy for both BD and MDD patients and for the distinction of control subjects from the BD and MDD patients, and the *p*-values of the permutation test were all less than 0.05. The implications of such findings are satisfactory, indicating that the classification model demonstrated a relatively encouraging utility at an individual level. This tool is expected to assist clinicians in making the most appropriate diagnosis for depressive patients, thereby surpassing the current symptom-based paradigm for diagnosis.

The present study has several limitations. Firstly, although our trained model achieved a relatively high accuracy of 90.3%, it should be noted that the model only focused on classifying BD and MDD. It is necessary to consider other clinical factors in practice. Secondly, the majority of the patients with BD and MDD received prior medication (different types, duration, and dosages) before MRI scanning, which potentially influenced the GM network. Future validation of our results should involve the recruitment of drug-naïve patients. Thirdly, the cross-sectional design hindered the direct understanding of GM network changes over the course of the diseases. Future studies with a longitudinal design are imperative. Fourthly, the choice of the brain parcellation template for constructing networks may have impacted the analysis results, emphasizing the need to consider different templates. Additionally, the physiological and behavioral implications of anatomic network alterations, derived from the construction of individual structural networks based on morphological distributions, require further understanding. Lastly, the accuracy of the machine learning models should be validated on larger samples or independent datasets to ensure generalizability.

In conclusion, notwithstanding the acknowledged limitations, our study revealed noteworthy global and nodal network alterations in individuals with BD and MDD when compared to healthy controls. Particularly, the BD group exhibited distinctive nodal parameter differences, primarily within the fronto-limbic circuit. The application of machine learning models utilizing structural brain parameters demonstrated an impressive 90.3% accuracy in distinguishing between individuals with BD and individuals with MDD. This research significantly contributes to advancing diagnostic precision in psychiatric disorders, providing valuable insights that will enhance clinical decision making in the differentiation between MDD and BD, while the successful application of machine learning models highlights a promising avenue for more accurate and objective diagnostic approaches, ultimately benefiting individuals with mood disorders and informing tailored treatment strategies.

## Figures and Tables

**Figure 1 diagnostics-14-00389-f001:**
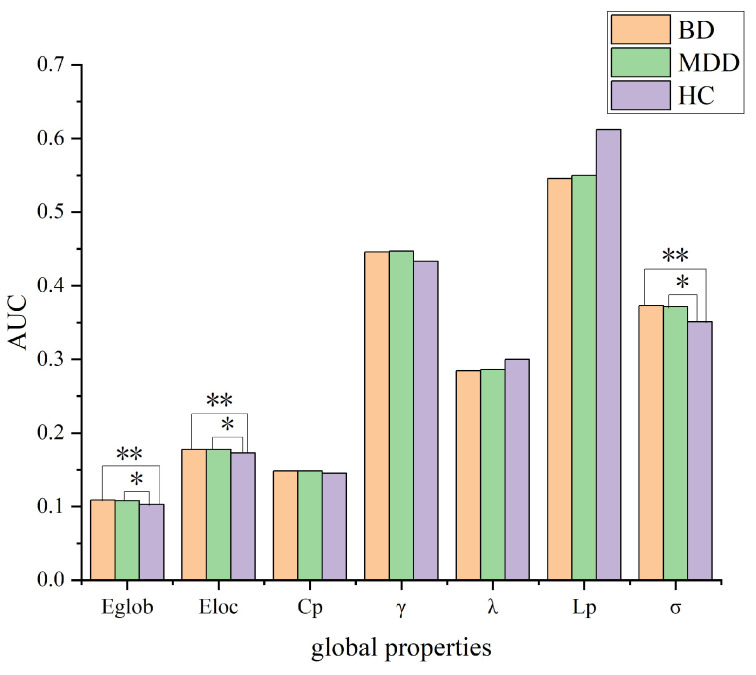
Graphs show differences in global topological properties among MDD, BD, and HC individuals. In the comparison between the HC and the BD groups, global efficiency (Eglob) (*p* = 0.022), local efficiency (Eloc) (*p* = 0.046) and sigma (σ) (*p* = 0.04) were significantly different. In the comparison between the HC and the MDD groups, global efficiency (Eglob) (*p* = 0.043), local efficiency (Eloc) (*p* = 0.030), and sigma (σ) (*p* = 0.014) were significantly different. An asterisk designates network metrics with a significant difference (*p* < 0.05). The symbol “*” indicates statistical significance between the MDD and the HC groups, while “**” indicates statistical significance between the BD and the HC groups.

**Figure 2 diagnostics-14-00389-f002:**
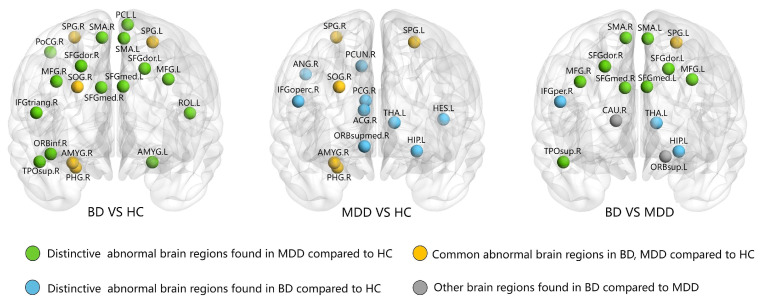
Graphs show the regions with significantly altered nodal centralities in the brain structural connectome among the HC, MDD, BD groups. The nodes were mapped onto the cortical surfaces by using the BrainNet Viewer package (http://www.nitrc.org/projects/bnv, accessed on 3 February 2024). Abbreviations: ACG = anterior cingulate and paracingulate gyri; AMYG = amygdala; ANG = angular gyrus; CAU = caudate nucleus; HES = Heschl gyrus; HIP = hippocampus; IFGoperc = inferior frontal gyrus, opercular part; IFGtriang = inferior frontal gyrus, triangular part; MFG = middle frontal gyrus; ORBinf = inferior frontal gyrus, orbital part; ORBsup = superior frontal gyrus, orbital part; ORBsupmed = superior frontal gyrus, medial orbital; PCG = posterior cingulate gyrus; PCL = paracentral lobule; PCUN = precuneus; PHG = parahippocampal gyrus; PoCG = postcentral gyrus; ROL = Rolandic operculum; SFGdor = superior frontal gyrus, dorsolateral; SFGmed = superior frontal gyrus, medial; SMA = supplementary motor area; SOG = superior occipital gyrus; SPG = superior parietal gyrus; THA = thalamus; TPOsup = temporal pole: superior temporal gyrus; R = right hemisphere; L = left hemisphere.

**Figure 3 diagnostics-14-00389-f003:**
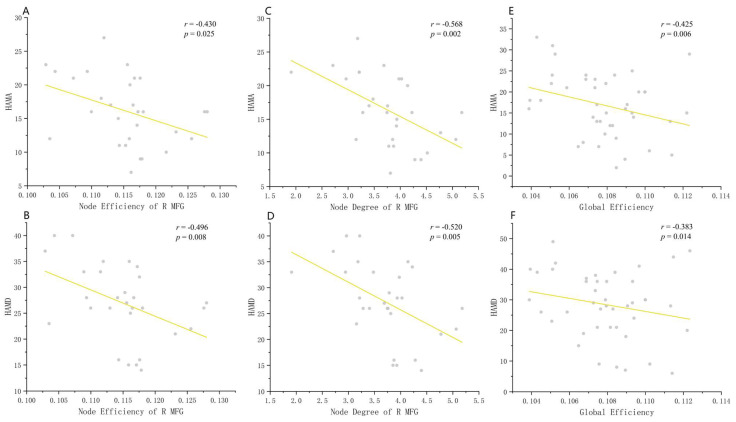
Graphs show the partial correlations between network metrics and clinical scales (HAMD and HAMA) for BD and MDD patients. (**A**,**B**) Node efficiency of the right middle frontal gyrus was significantly correlated with HAMA scores (*r* = −0.430, *p* = 0.025) and HAMD scores (*r* = −0.496, *p* = 0.008). (**C**,**D**) Nodal degree of the right middle frontal gyrus was significantly correlated with HAMA scores (*r* = −0.568, *p* = 0.002) and HAMD scores (*r* = −0.520, *p* = 0.005). (**E**,**F**) Global efficiency (E_glob_) was significantly correlated with HAMA scores (*r* = −0.425, *p* = 0.006) and HAMD scores (*r* = −0.383, *p* = 0.014). Abbreviations: MFG = middle frontal gyrus.

**Table 1 diagnostics-14-00389-t001:** Demographic characteristics and clinical scales.

Variables				*p* Value		
	BD (*n* = 32)	MDD (*n* = 43)	HC (*n* = 44)	BD vs. MDD	BD vs. HC	MDD vs. HC
Age (years)	20.28 ± 11.07	22.37 ± 13.92	21.95 ± 8.181	0.430 ^a^	0.525 ^a^	0.864 ^a^
Sex (M/F)	6/26	10/33	11/33	0.778	0.586	1.000
HAMD	26.88 ± 8.50	28.40 ± 10.96	NA	0.517	NA	NA
HAMA	17.28 ± 6.38	17.40 ± 8.29	NA	0.948	NA	NA
YMRS	26.923 ± 8.026	NA	NA	NA	NA	NA

^a^: The differences in age among the three groups were assessed using the least-significant difference method in the analysis of variance. Abbreviations: HAMD: Hamilton Depression Scale; HAMA, Hamilton Anxiety Scale; YMRS: Young Mania Rating Scale.

**Table 2 diagnostics-14-00389-t002:** Regions showing differences in nodal centralities in the bipolar, major depressive disorder, and health control groups.

Brain Regions	*p* Values		
	Nodal Degree	Nodal Efficiency	Nodal Betweenness
BD vs. MDD			
L Superior frontal gyrus, dorsolateral	**0.036** ** ↑ **	**0.016** ** ↑ **	0.266
R Superior frontal gyrus, dorsolateral	**0.013** ** ↑ **	**0.011** ** ↑ **	0.342
L Superior frontal gyrus, orbital part	**0.036** ** ↓ **	**0.039** ** ↓ **	0.358
L Middle frontal gyrus	0.278	**0.034** ** ↑ **	0.358
R Middle frontal gyrus	**0.015** ** ↑ **	0.056	0.406
R Inferior frontal gyrus, opercular part	**0.045** ** ↓ **	0.068	**0.044** ** ↓ **
L Supplementary motor area	**0.009** ** ↑ **	**0.025** ** ↑ **	0.138
R Supplementary motor area	**0.013** ** ↑ **	**0.011** ** ↑ **	0.063
L Superior frontal gyrus, medial	0.131	**0.048** ** ↑ **	0.477
R Superior frontal gyrus, medial	**0.034** ** ↑ **	**0.034** ** ↑ **	0.209
L Hippocampus	**0.020** ** ↓ **	**0.034** ** ↓ **	0.406
L Superior parietal gyrus	**0.034** ** ↓ **	0.117	**0.013** ** ↓ **
R Caudate nucleus	0.406	0.406	**0.009** ** ↑ **
L Thalamus	0.152	0.214	**0.009** ** ↑ **
R Temporal pole: superior temporal gyrus	**0.034** ** ↓ **	0.068	0.117
BD vs. HC			
L Superior frontal gyrus, dorsolateral	0.329	**0.004** ** ↑ **	0.165
R Superior frontal gyrus, dorsolateral	0.126	**0.003** ** ↑ **	0.471
L Middle frontal gyrus	0.433	**0.025** ** ↑ **	0.185
R Middle frontal gyrus	0.140	**0.018** ** ↑ **	0.330
R Inferior frontal gyrus, triangular part	0.266	**0.006** ** ↑ **	0.471
R Inferior frontal gyrus, orbital part	**0.017** ** ↑ **	**0.006** ** ↑ **	**0.010** ** ↑ **
L Rolandic operculum	0.117	0.318	**0.029** ** ↓ **
L Supplementary motor area	**0.004** ** ↑ **	**0.003** ** ↑ **	0.471
R Supplementary motor area	**0.003** ** ↑ **	**0.003** ** ↑ **	0.179
L Superior frontal gyrus, medial	0.163	**0.003** ** ↑ **	0.480
R Superior frontal gyrus, medial	**0.017** ** ↑ **	**0.004** ** ↑ **	0.280
R Parahippocampal gyrus	0.090	**0.029** ** ↑ **	0.228
L Amygdala	0.105	**0.038** ** ↑ **	0.459
R Amygdala	0.110	**0.043** ** ↑ **	0.239
R Superior occipital gyrus	**0.008** ** ↑ **	**0.003** ** ↑ **	**0.016** ** ↑ **
R Postcentral gyrus	0.290	**0.039** ** ↑ **	**0.013** ** ↑ **
L Superior parietal gyrus	**0.003**	**0.008** ** ↓ **	**0.003** ** ↓ **
R Superior parietal gyrus	**0.017** ** ↓ **	0.188	**0.003** ** ↓ **
L Paracentral lobule	0.464	0.244	**0.029** ** ↓ **
R Temporal pole: superior temporal gyrus	**0.029** ** ↓ **	0.348	0.224
MDD vs. HC			
R Inferior frontal gyrus, opercular part	0.207	**0.049** ** ↑ **	0.333
R Superior frontal gyrus, medial orbital	0.082	**0.037** ** ↑ **	0.432
R Anterior cingulate and paracingulate gyri	0.058	**0.037** ** ↑ **	0.055
R Posterior cingulate gyrus	**0.037** ** ↓ **	0.058	**0.037** ** ↓ **
L Hippocampus	**0.043** ** ↑ **	**0.014** ** ↑ **	0.499
R Parahippocampal gyrus	0.051	**0.042** ** ↑ **	0.166
R Amygdala	**0.043** ** ↑ **	0.064	0.180
R Superior occipital gyrus	0.064	**0.037** ** ↑ **	0.247
L Superior parietal gyrus	0.172	0.256	**0.042** ** ↓ **
R Superior parietal gyrus	0.172	0.282	**0.027** ** ↓ **
R Angular gyrus	0.381	0.229	**0.037** ** ↑ **
R Precuneus	0.350	0.198	**0.037** ** ↑ **
L Thalamus	0.198	0.291	**0.014** ** ↓ **
L Heschl gyrus	0.055	**0.037↑**	0.400

All brain regions identified by automated anatomical labeling (AAL). ↑, in BD vs. MDD, the bipolar disorder group showed greater nodal centralities than the major depressive disorder group; in BD vs. HC, the bipolar disorder group showed greater nodal centralities than the healthy control group; in MDD vs. HC, the major depressive disorder group showed greater nodal centralities than the healthy control group. ↓, in BD vs. MDD, the bipolar disorder group showed lower nodal centralities than the major depressive disorder group; in BD vs. HC, the bipolar disorder group showed lower nodal centralities than the healthy control group; in MDD vs. HC, the major depressive disorder group showed lower nodal centralities than the healthy control group. Bold values indicate *p* < 0.05.

**Table 3 diagnostics-14-00389-t003:** Classification performance of commonly used machine learning models for structural connectivity matrix and sub-matrix of the structural connectivity matrix.

Model Performance	BAC (%)	SEN (%)	SPE (%)	*p* Value ^a^
HC vs. BD + MDD (structural connectivity matrix (90 × 90))				
SVM	75.1%	80.5%	69.8%	<0.001
Xgboost	91.1%	99.0%	83.3%	<0.001
RF	87.9%	98.3%	77.4%	<0.001
AE + SVM	67.0%	68.5%	65.6%	0.014
AE + Xgboost	86.73	97.0%	76.5%	<0.001
AE + RF	85.7%	98.2%	73.2%	<0.001
DNN + SVM	63.5%	63.8%	63.0%	0.025
DNN + Xgboost	83.2%	98.3%	68.0%	<0.001
DNN + RF	83.5%	99.0%	68.1%	<0.001
BD vs. MDD (structural connectivity matrix (90 × 90))				
SVM	78.8%	91.0%	66.7%	<0.001
Xgboost	85.0%	98.0%	72.0%	<0.001
RF	83.7%	98.0%	69.3%	<0.001
AE + SVM	79.2%	90.3%	68.0%	<0.001
AE + Xgboost	79.7%	98.0%	61.3%	<0.001
AE + RF	80.2%	97.7%	62.7%	<0.001
DNN + SVM	57.3%	57.3%	57.3%	0.004
DNN + RF	78.0%	98.7%	57.3%	<0.001
DNN + Xgboost	76.0%	98.7%	43.3%	<0.001
BD vs. MDD (sub-matrix of structural connectivity matrix (15 × 15))				
SVM	81.0%	91.3%	70.7%	<0.001
Xgboost	87.0%	98.0%	76.0%	<0.001
RF	84.3%	98.0%	70.7%	<0.001
AE + SVM	73.8%	78.3%	69.3%	<0.001
AE + Xgboost	83.0%	98.0%	68.0%	<0.001
AE + RF	85.5%	97.7%	73.3%	<0.001
DNN + SVM	86.8%	87.0%	86.7%	<0.001
DNN + Xgboost	90.3%	98.0%	82.7%	<0.001
DNN + RF	83.3%	98.7%	68.0%	<0.001

^a^: The *p*-value obtained from conducting 10,000 permutation tests. Abbreviations: BAC, balanced accuracy; SEN, sensitivity; SPE, specificity; SVM, support vector machine; Xgboost, eXtreme gradient boosting; RF, random forest; AE, autoencoder; DNN, deep neural networks.

## Data Availability

The data presented in this study are available on request from the corresponding author.

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
