# Peer review of "Innovative Neuroimaging Biomarker Distinction of Major Depressive Disorder and Bipolar Disorder through Structural Connectome Analysis and Machine Learning Models"

_diagnostics, 2024, doi:10.3390/diagnostics14040389_

Round 1

Reviewer 1 Report

Comments and Suggestions for Authors

The research focuses on the structural analysis of the brain connectome in MDD and BD. The study has a number of limitations that reduce its quality.

1. It is confusing to suggest that BD is difficult to distinguish from only MDD. The differential diagnosis for bipolar disorder can also be schizophrenia. The fact that MDD is not the only diagnosis that can be misdiagnosed in patients with bipolar disorder should be mentioned in the introduction. This should also be mentioned in the limitations of the study. 

The introduction also lacks information about the brain connectome and its significance in the development of bipolar disorder and depression. 

2. Figure 3 is not referenced in the text.  Although the significance level is less than 0.05 the correlation coefficients show only a moderate but not strong association between the traits. This should be mentioned. 

3. Paragraph 3.4.- authors should add the values of correlation coefficients and probability. 

4. Paragraph 3.5. Why did the authors not aim to separate control from BD and MDD using machine learning? What is the specificity of this method in separating control from pathology? Understanding how the model separates healthy people from individuals with pathology provides information about its future applicability. 

5. Lines 386-390 should be moved to the introduction. "However, more than 60% of BD patients will still be misdiagnosed" - this fact is mentioned for the second time in the manuscript.

Author Response

1.It is confusing to suggest that BD is difficult to distinguish from only MDD. The differential diagnosis for bipolar disorder can also be schizophrenia. The fact that MDD is not the only diagnosis that can be misdiagnosed in patients with bipolar disorder should be mentioned in the introduction. This should also be mentioned in the limitations of the study. 

Response: We have added other potential misdiagnoses of BD in the introduction, and discussed this situation in the limitations.

“ The clinical symptoms of bipolar disorder (BD) overlap with other psychiatric disorders, such as major depressive disorder (MDD), schizophrenia(SP), attention deficit hyperactivity disorder (ADHD) [2], which can lead to misdiagnosis. Among them, MDD is the most common misdiagnosis for BD. ”

“ Firstly, although our trained model achieved a relatively high accuracy of 90.3%, it should be noted that the model only focuses on classifying BD and MDD. It is necessary to consider other clinical factors in practice. ”

The introduction also lacks information about the brain connectome and its significance in the development of bipolar disorder and depression. 

Response: We have added the information of the brain connectome to the development of BD and MDD in the introduction.

“ For example, recent reports have highlighted the identification of potential causes for the alteration of connectome structure in BD [19-20] and MDD [21-22] through graph theory analysis. ”

2.Figure 3 is not referenced in the text.  Although the significance level is less than 0.05 the correlation coefficients show only a moderate but not strong association between the traits. This should be mentioned.

Response: Figure 3 has been referenced in Paragraph 3.4. Meanwhile, in the discussion section, the strength of the correlation was also discussed.

“ The absolute value of the correlation coefficient is only moderate, but it still holds certain clinical significance. ”

  1. Paragraph 3.4.- authors should add the values of correlation coefficients and probability. 

Response: The values of correlation coefficients and probabilities have been added to their respective positions in Paragraph 3.4.

“ the nodal efficiency (HAMD scores, r = -0.496, p = 0.008, FDR corrected; HAMA scores, r = -0.430, p = 0.025, FDR corrected) and degree (HAMD scores, r = -0.520, p = 0.005, FDR corrected; HAMA scores, r = -0.568, p = 0.002, FDR corrected) of the right middle frontal gyrus in the BD (Figure 3). ”

“ ......MDD, HAMD scores (r = -0.425, p = 0.006, FDR corrected) and HAMA scores (r = -0.383, p = 0.014, FDR corrected) ”

3.Paragraph 3.5. Why did the authors not aim to separate control from BD and MDD using machine learning? What is the specificity of this method in separating control from pathology? Understanding how the model separates healthy people from individuals with pathology provides information about its future applicability. 

Response: We have added the classification of HC from BD, and MDD in the machine learning section and we have added relevant content to the methods, results, and discussion sections. Currently, the differential diagnosis of BD and MDD is primarily based on questionnaire-based symptom assessments, which are relatively subjective. In contrast, our study utilizes imaging-based approaches, providing a more objective assessment. Although our trained model cannot be directly applied to clinical settings for distinguishing between BD and MDD, it can serve as an adjunct tool to assist clinicians in diagnosis. Simultaneously, machine learning can integrate imaging indicators with other metrics, such as genetic markers, to train models that assist clinical diagnosis from multiple perspectives.

“ We attempted to differentiate control subjects from those with BD and MDD. Each subject's GM morphological matrices (90 × 90 Pearson correlation matrix) were used as features, while BD and MDD were assigned the same label of 1, HC were labeled as 0...... ”

“ ...... We trained SVM, RF, and Xgboost models. Additionally, we employed AE and DNN for feature reduction, and subsequently trained the aforementioned three conventional machine learning models using the reduced features. ”

“ Utilizing GM morphological matrices (90 × 90) as features to differentiate HC from those with BD and MDD, the mean balanced accuracy for classification using SVM is 75.1%, 87.9% for RF, and 91.1% for Xgboost. Autoencoder is applied as the first-stage dimensionality reduction technique, leading to mean balanced accuracy percentages of 67.0% for SVM, 85.7% for RF, and 86.7% for Xgboost in the second stage. When DNN is utilized as the first-stage dimensionality reduction, the mean balanced accuracy for SVM in the second stage is 63.5%, 83.5% for RF, and 83.2% for Xgboost (Table 3). ”

“However, in the classification of control subjects from those with BD and MDD, we found that Xgboost without dimensionality reduction achieved the highest classification accuracy (91.1%), indicating that the selection of machine learning models should be based on the actual data situation. Nevertheless, it is worth noting that we obtained a relatively high classification accuracy for both the classification of BD and MDD and the classification of control subjects from BD and MDD, and the P-values from the permutation test were all less than 0.05.”

  1. Lines 386-390 should be moved to the introduction. "However, more than 60% of BD patients will still be misdiagnosed" - this fact is mentioned for the second time in the manuscript.

Response: The sentence and the cited references have been moved to introduction.

“ Previous studies have indicated that as many as 69% of BD patients have been misdiagnosed [5-7]. ”

Reviewer 2 Report

Comments and Suggestions for Authors

The authors attempt to differentiate between major depressive disorder and bipolar mood disorder by combining structural neuroimaging, graph theory and machine learning in an innovative study.

Major concerns

1. Medication is a major confounding variable. Only eight patients were drug-naive at the time of assessment. Have the authors made any attempt to take medication status, including type of medication, duration of medication and dosage (e.g. chlorpromazine equivalents), into account statistically or through some form of modelling?

2. In Figure 3, it seems to me that in both graphs A and B the regression line is unduly biased by outliers. I recommend that the authors consider re-analysing the corresponding data without these outliers.

Minor points

3. Line 222: the authors use the term "sex" (correctly, in my view). However, in line 196 and in Table 1 the authors use the term "gender" (incorrectly, in my view). Please harmonise these.

4. Line 251: if the authors are going to capitalise "Heschl", then in this line they should also capitalise "rolandic" to "Rolandic".

5. Table 3: please state the measure of dispersion for the first row of data.

6. Line 202: please state the names of the individual R packages used (and perhaps give the corresponding references, if readily available).

Comments on the Quality of English Language

The authors inappropriately use initial capitalisations in many places.

Author Response

1.Medication is a major confounding variable. Only eight patients were drug-naive at the time of assessment. Have the authors made any attempt to take medication status, including type of medication, duration of medication and dosage (e.g. chlorpromazine equivalents), into account statistically or through some form of modelling?

Response: Considering practicality and feasibility in real situations, we have controlled for medication status(no medication use, the use of antidepressants alone, he combined use of antidepressants and antipsychotic drugs) in our analysis. While we employed the Gaussian process regression method along with a kernel function to control for medication status.  Then, we also employed a random permutation test (n = 10000) on the model used and data standardization and normalization to train the machine learning model to mitigate the impact of confounding variables on the experiment.

“ To mitigate the confounding effect of medication on the machine learning models differentiating between BD and MDD, we coded no medication use as 1, the use of antidepressants alone as 2, and the combined use of antidepressants and antipsychotic drugs as 3. Because of the non-linear relationship between features and labels, we employed the Gaussian process regression method along with a kernel function to control for medication status. Specifically, a regression model was established for each feature to capture how it varied with medication status, and the predicted feature values were subtracted from the actual feature values to obtain residuals. These residuals were subsequently used as features in the analysis. Additionally, features were standardized and normalized after controlling for confounding variables. ”

“ By shuffling the data labels 10000 times(for instance, exchanging labels of some data in a binary classification problem), retraining the model, and evaluating its performance, we can assess whether the model's predictions significantly surpass random guessing. ”

  1. In Figure 3, it seems to me that in both graphs A and B the regression line is unduly biased by outliers. I recommend that the authors consider re-analysing the corresponding data without these outliers.

Response: Outliers were detected in A, B, C, and D through box plot analysis. After the removal of these outliers, a partial correlation analysis was re-conducted using the data with outliers excluded, and redrawn the Figure 3.

  1. Line 222: the authors use the term "sex" (correctly, in my view). However, in line 196 and in Table 1 the authors use the term "gender" (incorrectly, in my view). Please harmonise these.

 Response: The term "gender" in Table 1 has been changed to "sex".

  1. Line 251: if the authors are going to capitalise "Heschl", then in this line they should also capitalise "rolandic" to "Rolandic".

Response: We have changed "rolandic" to "Rolandic".

  1. Table 3: please state the measure of dispersion for the first row of data.

Response: We have added relevant details regarding the model evaluation metrics in the machine learning section.

“ All models were trained and evaluated using 5-fold stratified cross-validation. For each iteration of cross-validation, a subset of the dataset was used to evaluate the model (i.e., the test set) while the remaining four subsets were used for training (i.e., the training set). In the dimensionality reduction stage, we only performed dimensionality reduction based on the training set, and the test set was solely used for performance evaluation, never for model tuning or training. Model performance was determined by balanced accuracy, sensitivity, and specificity. ”

  1. Line 202: please state the names of the individual R packages used (and perhaps give the corresponding references, if readily available).

Response: The names of the relevant R packages and their corresponding website links have been added.

“ Statistical analysis was conducted using R software (version 4.1.2), and the R package "ppcor" (https://cran.r-project.org/web/packages/ppcor/index.html) was employed for partial correlation analysis. ”

Round 2

Reviewer 2 Report

Comments and Suggestions for Authors

I thank the authors for addressing my previous comments.

Comments on the Quality of English Language

Minor amendments may be made during the proof reading stage.